# Using the Turnover Time Index to Identify Potential Strategic Groundwater Resources to Manage Droughts within Continental Spain

**David Pulido-Velazquez [1,*], Javier Romero [2], Antonio-Juan Collados-Lara [1], Francisco J. Alcalá [3,4], Francisca Fernández-Chacón [5] and Leticia Baena-Ruiz [1]**

[1]   Instituto Geológico y Minero de España, Urb. Alcázar del Genil, 4. Edificio Zulema, Bajo, 18006 Granada, Spain; aj.collados@igme.es (A.-J.C.-L.); l.baena@igme.es (L.B.-R.)

[2]   Campus de los Jerónimos s/n, Universidad Católica San Antonio de Murcia, Guadalupe, 30107 Murcia, Spain; romerog.javier@gmail.com

[3]   Instituto Geológico y Minero de España, Ríos Rosas, 23, 28003 Madrid, Spain; fj.alcala@igme.es

[4]   Instituto de Ciencias Químicas Aplicadas, Facultad de Ingeniería, Universidad Autónoma de Chile, Santiago 7500138, Chile

[5]   IES Ribera de Fardes, Cerro de los Almendrillos, Purullena, 18519 Granada, Spain; paquifchacon@gmail.com

*   Correspondence: d.pulido@igme.es; Tel.: +34-(95)-818-3143

**Abstract:** The management of droughts is a challenging issue, especially in water scarcity areas, where this problem will be exacerbated in the future. The aim of this paper is to identify potential groundwater (GW) bodies with reduced vulnerability to pumping, which can be used as buffer values to define sustainable conjunctive use management during droughts. Assuming that the long term natural mean reserves are maintained, a preliminary assessment of GW vulnerability can be obtained by using the natural turnover time (T) index, defined in each GW body as the storage capacity (S) divided by the recharge (R). Aquifers where R is close to S are extremely vulnerable to exploitation. This approach will be applied in the 146 Spanish GW bodies at risk of not achieving the Water Framework Directive (WFD objectives, to maintain a good quantitative status. The analyses will be focused on the impacts of the climate drivers on the mean T value for Historical and potential future scenarios, assuming that the Land Use and Land Cover (LULC) changes and the management strategies will allow maintenance of the long term mean natural GW body reserves. Around 26.9% of these GW bodies show low vulnerability to pumping, when viewing historical T values over 100 years, this percentage growing to 33.1% in near future horizon values (until 2045). The results show a significant heterogeneity. The range of variability for the historical T values is around 3700 years, which also increases in the near future to 4200 years. These T indices will change in future horizons, and, therefore, the potential of GW resources to undergo sustainable strategies to adapt to climate change will also change accordingly, making it necessary to apply adaptive management strategies.

**Keywords:** drought; vulnerability to pumping; residence time; conjunctive use; sustainable management; climate change; adaptation strategies; Spanish GW bodies in quantitative risk

## 1. Introduction

The management of droughts is a challenging issue, especially in water scarcity areas with water deficits in terms of long-term average conditions [1]. These deficits can be observed in precipitation, soil moisture, river discharge or supply in relation to water demands, defining respectively different drought types: meteorological, agricultural, hydrological, or even operational droughts depending on the variables used to assess them [2]. For example, operational drought or scarcity [3] is related to the

deficit in water demand satisfaction in a system. In most of these areas, the frequency and intensity of the drought events will be exacerbated in the future, due to climate change [4]. In this framework, groundwater (GW) may play a significant role for sustainable management of water scarcity, due to its role as buffer value, providing additional resources that can be temporarily employed to cover necessities during critical droughts [5]. GW resources are also crucial for appropriate analyses of scarcity, and due to aquifer status have an important influence on water availability to fulfil demands. GW overexploitation is an issue with even higher impacts in lowering GW levels than climate change in many regions, especially in the Mediterranean area [6]. These impacts have been explicitly analysed and discussed in research works on coastal [7] and non-coastal Mediterranean aquifers [8]. Drought also exacerbates aquifer overexploitation, a significant issue in the Mediterranean area [9].

Increased water availability has resulted in an even larger increase in water supply demands, as demonstrated in Tunisia by Besbes et al. [10]. Groundwater that was frequently exploited as a consequence of the silent revolution in agricultural development gave rise to a demand for irrigation which accounts for 60–80% of the total water demand in the Mediterranean area today. As a result of this highly exacerbated groundwater use, water tables in some regions have fallen by as much as several hundred meters, such as in southeast Spain [11]. Among other significant immediate effects, seawater intrusion into coastal aquifers has also been recorded [12]. Notwithstanding the numerous interpretations of overexploitation as a general term [13], it is hereinafter used to describe the exhaustive use of groundwater resources that pose a subsequent risk to the preservation of groundwater quality and quantity in the long-term, and/or to additional services provided by the aquifers.

On the other hand, the legal EU water management context defined by the Water Framework Directive [14] aims to achieve a sustainable management of the resources, maintaining a good status of surface and GW bodies. The state members, for the different planning horizons, have identified Water Bodies at risk of not achieving the Water Framework Directive (WFD) (2000) [14] objectives and have proposed programs oriented to fulfill these targets. Therefore, in the decision-making process when managing water resource systems, special attention should be paid to the management of these GW bodies.

The concept of vulnerability is closely related to GW body status and risks. It has been extensively studied from the perspective of vulnerability to surface pollution. Different approaches and techniques [15] can be applied to assess both intrinsic and specific vulnerability. Intrinsic vulnerability focuses on analyses of the ease with which any surface pollution can reach and extend within the aquifer saturated zone [16]. Specific vulnerability refers to a particular contaminant or groups of contaminants, taking into account their properties and the potential processes and interactions that may influence them [17]. From this qualitative point of view, the vulnerability of a GW-resource to pollution depends on intrinsic susceptibility, which depends on the aquifer properties, the associated sources of water, the distribution and types of contamination sources (natural and/or anthropogenic), and the transport of the contaminants [18]. The most commonly employed methods to assess vulnerability are the index-based approaches [8] that include methods such as DRASTIC [19], Aquifer vulnerability index (AVI) [20] and SINTACS [21] or adaptation of them. In the Mediterranean region, SINTACS and SINTACS-derived methods are the most commonly applied, both for intrinsic and specific vulnerability assessment [22]. This GW vulnerability concept has also been linked to variable GW ages, travel and residence times.

GW age can be estimated by using environmental and isotope tracers [23,24] and model simulations [25,26]. It has been related to vulnerability of production wells to contamination in some cases [27]. The mean age of the water leaving the system or the mean residence time is also known as the mean GW age or renewal time, and can be approached, under natural conditions, by the mean natural GW turnover time (T) index, which is obtained by dividing the total storage capacity (S) by the net GW recharge (R) [23]. Therefore, the value of this index in the future could be affected by climate change scenarios and their impacts on rainfall aquifer recharge [28–30].

In this paper we propose a novel method to perform a preliminary analysis of GW vulnerability to intensive pumping during drought periods, assuming that the long term natural mean reserves are maintained by the actual recharge of the main inflow of groundwater resources. This will be analyzed under historical and potential future climate scenarios. The pumping vulnerability concept is introduced and assessed by applying a T index approach. This allows us to identify potential strategic GW bodies for sustainable conjunctive use management of critical droughts in water scarcity areas in continental Spain. We also studied the significance and variability of the R and S variables employed to obtain the T index depending on the aquifer lithology. Finally, we analyzed whether some potential explanatory variables could be employed to describe the T distribution. Due to T dependencies on R, which can be estimated from the effective precipitation (precipitation minus the actual evapotranspiration), by applying an effective recharge coefficient (C), the explanatory analyses were also extended to the recharge coefficient.

## 2. Materials and Methods

### 2.1. Methodology

This novel method intends to perform a preliminary analysis of GW vulnerability to intensive pumping during drought periods through the renewal time of resources (GW age), approached by the T index as the S/R ratio. Assuming that the long term natural mean reserves are kept invariant and the actual recharge is the main inflow of groundwater resources, the GW bodies with high renewal time will be less vulnerable to pumping than those with low values, even in periods in which pumping is smaller than mean R. This can be especially relevant in Basins or Water Resource systems with scarce reserves where long and intensive droughts appear and will be exacerbated in the future due to climate change. The methodology is summarized in Figure 1.

Making a parallel between unconfined aquifers and reservoirs, the GW discharge (Q) will start when the potential aquifer storage reaches the threshold level of the surface connection (see Figure 1). Assuming that there is no pumping, a preliminary assessment of the natural mean age of the groundwater leaving the GB body through the connection with the surface system (springs and or stream-aquifer interaction boundary conditions) can be obtained through the natural mean T index, defined as:

$$T = S/R \tag{1}$$

where T, S, and R are defined in the caption to Figure 1.

In each GW body, S can be obtained by combining information about the geometry and the storage coefficients, which can be derived from different sources (e.g., field works, models and/or research papers and official reports, as well as the River Basin Plans published by the different River Basin authorities). The historical R can be estimated through field work or previously calibrated models [29,31,32]. The historical mean T value can be estimated by combining the mean historical R values with S in accordance with Equation (1).

The impacts of future potential climatic scenarios on GW bodies R, and, therefore, on their T index, requires climatic scenarios to be downscaled and propagated with a previously calibrated recharge model. In order to generate future local scenarios from the simulations of the last future emission scenarios defined by the Intergovernmental Panel on Climate Change (IPCC) (AR5) with climatic models, we need to correct these with local historical climatic data for the case study [33]. Two different statistical approaches may be applied to perform the correction or downscaling, a delta change approach and/or a bias correction approach [34]. Local scenarios can be generated for a future horizon or a specific global warming level (e.g., 1 or 2 °C above the reference period) [35]. We propose to define an equi-feasible ensemble of multiple local projections, which are supposed to produce more robust and representative scenarios than those based on single projections [36]. The impacts of the generated potential local scenarios on the mean T will require future mean R to be estimated, which will

be assessed by propagating/simulating the generated climatic scenarios with previously calibrated recharge models.

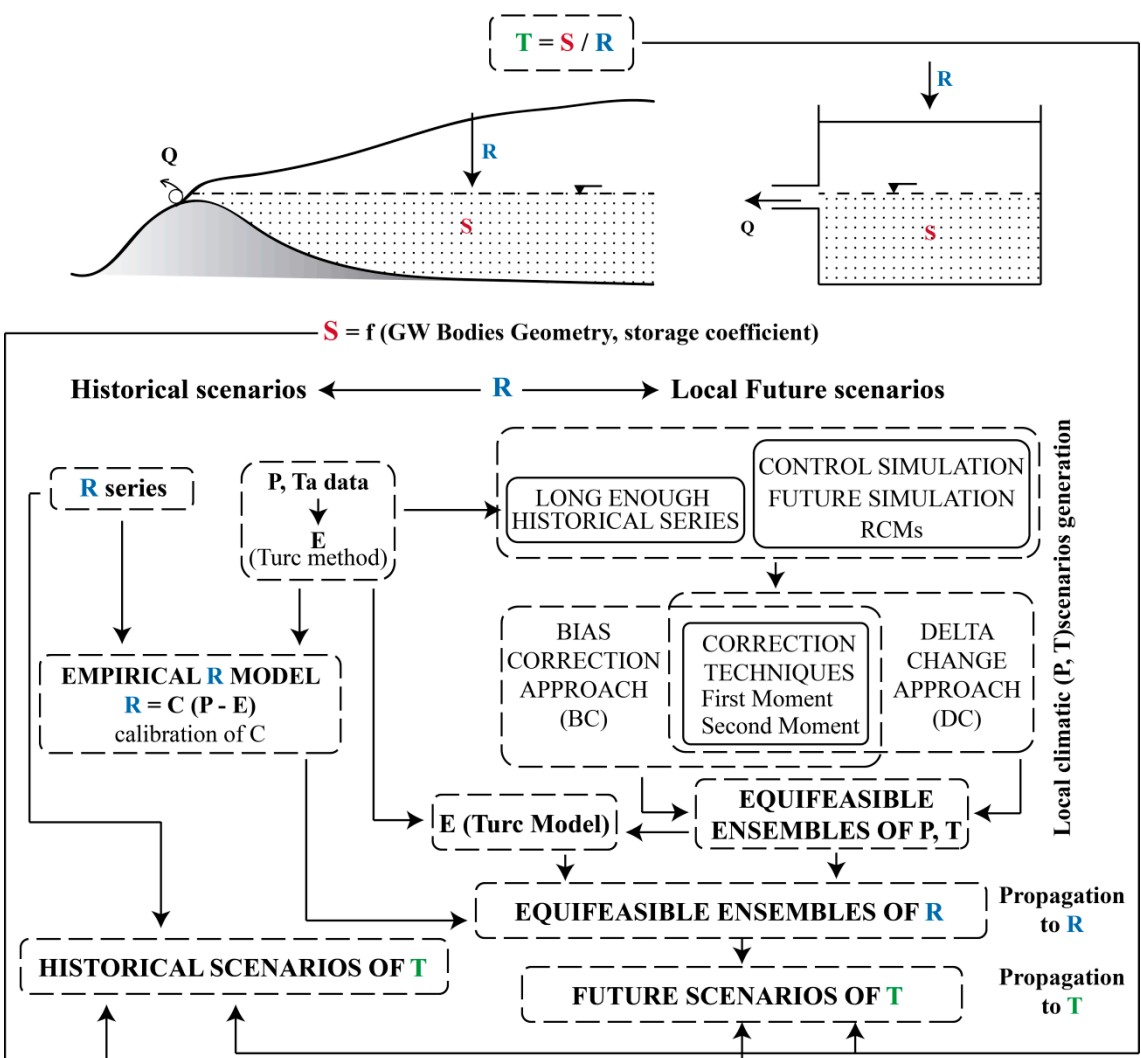

**Figure 1.** Flowchart of the methodology developed to assess groundwater (GW) bodies' vulnerability to pumping. Notation and units for variables used: P, E, R, and Q are respectively precipitation, actual evapotranspiration, net GW recharge from P, and net GW discharge in mm year$^{-1}$; Ta is temperature in °C; C and S are respectively a dimensionless effective recharge coefficient (−) and a GW storage (Mm$^3$); and T is the natural turnover time index in years.

## 2.2. Materials: Description of the Study Area and the Available Information

### 2.2.1. Location, Geological Context and Historical Climatic Data

In continental Spain, 717 GW bodies were defined for Water Planning. These GW bodies cover 71% of the territory. The definition of sustainable management strategies for water resource systems should pay special attention to water bodies at risk of not achieving the WFD (2000) objectives. For this reason, in this study we focused on the 146 Spanish GW bodies at risk of not fulfilling the WFD (2000) objectives (see Figure 2) in order to identify which could be potentially considered strategic for a sustainable conjunctive use management of droughts.

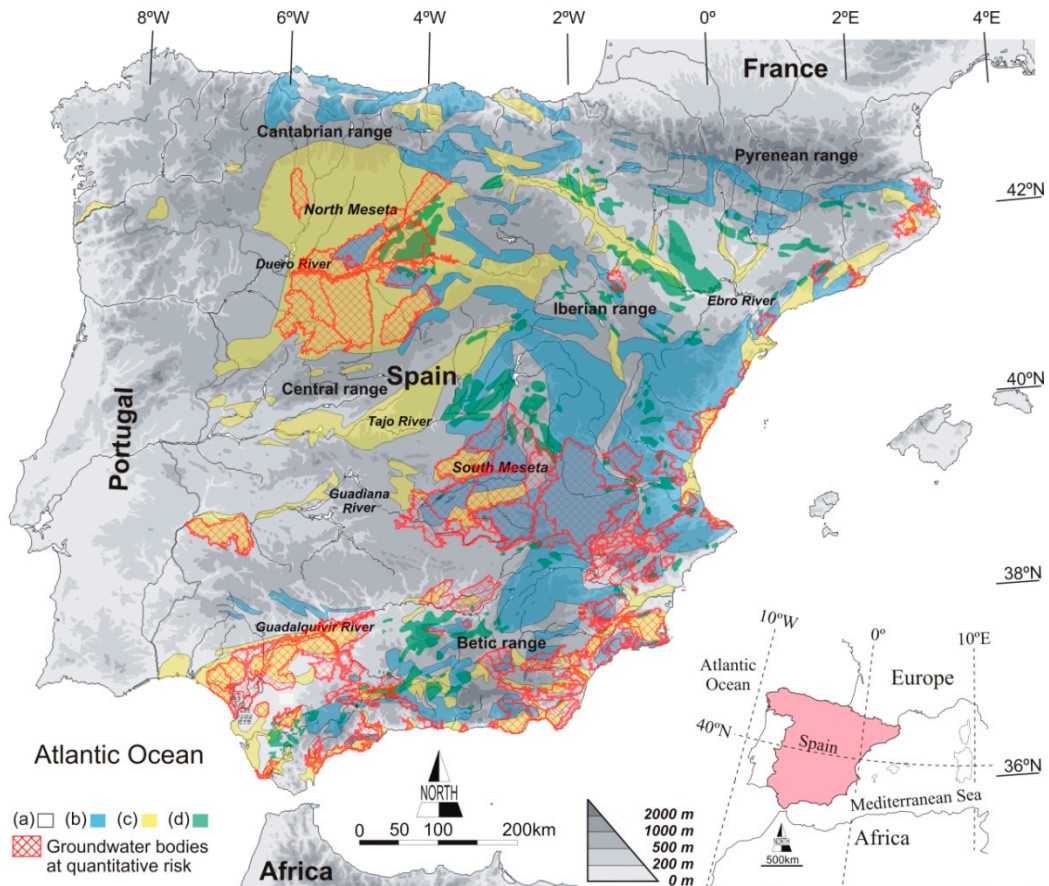

**Figure 2.** Map of continental Spain, showing the 146 Spanish GW bodies at quantitative risk of not fulfilling the Water Framework Directive (WFD) [14] (2000) objectives (red shadowed areas), the main mountain ranges and hydrographic basins, and the hydrogeological behavior of geological materials forming the GW bodies according to permeability type [37], modified from [31] as: (**a**) low to moderate permeability pre-Triassic metamorphic rocks, granitic outcrops, and Triassic to Miocene marly sedimentary formations; (**b**) moderate to high permeability Paleozoic to Tertiary; (**c**) moderate to high permeability Pleo-Quaternary detritic; and (**d**) Triassic to Miocene evaporitic outcrops.

After the WFD [1] came into effect, the European Environment Agency established guidelines for declaring those GW bodies at risk of not fulfilling a good quantitative and qualitative level in the 2020 horizon, as well as general measures to mitigate negative impacts. Declaration of GW bodies at quantitative risk was based on particular net GW balances resulting from GW extractions and losses (pumping, direct evaporation, net GW discharge, lateral outflows) and the available renewable resources (net GW recharge, irrigation and urban returns, stream losses, and lateral inflows). The exploitation index in each GW body, defined as extractions (pumping) divided by the available renewable resources plus the environmental flow, was proposed as a measure of sustainability; 1 was considered the minimum threshold below which there is a GW imbalance. A GW body is classified as having a bad quantitative status when the exploitation index is above 0.8 and there is a clear piezometric level depletion trend over a large fraction of its surface. These GW bodies cover 16% of continental Spain.

The varied geology of continental Spain determines many relatively small high-yielding GW bodies widely distributed throughout its territory. The most important GW bodies are in Pleo-Quaternary sedimentary formations and Triassic to Tertiary carbonate massifs (Figure 2). The former consists of inland GW bodies surrounded by mountain ranges, small alluvial and piedmont units, and deltaic formations on infilled estuaries in coastal areas. Carbonate massifs are common in quite extensive but compartmentalized areas along the northern, eastern, and southern ranges [37]. To a minor

extent, the weathered and fissured granite and Paleozoic shale formations in northern, southern, and north-eastern ranges, contain small aquifers of local significance not catalogued as GW bodies. Attending to hydrogeological behavior of geological materials forming the GW bodies deduced from the permeability type, Alcalá and Custodio [31] classified the geological materials forming the GW bodies as: (a) low to moderate permeability pre-Triassic metamorphic rocks, granitic outcrops, and Triassic to Miocene marly sedimentary formations forming low productive GW bodies and impervious areas; (b) moderate to high permeability Paleozoic to Tertiary carbonates forming mostly highly productive GW bodies; (c) moderate to high permeability Pleo-Quaternary detrital materials corresponding typically to moderately to highly productive GW bodies; and (d) Triassic to Miocene evaporitic outcrops characterizing areas subjected to potential GW pollution due to natural sources of salinity (Figure 2).

For the purposes of this research, historical climatic (temperature and precipitation) data collected from the Spain02 project [38] for the chosen reference period (1976–2005) were used. This precipitation data includes both rainfall and snowfall. Temperature and precipitation show significant spatial heterogeneity as a result of highly variable climatic conditions. Annual mean P ranges from 190 mm year$^{-1}$ in south-eastern semiarid regions to over 2000 mm year$^{-1}$ in humid northern locations (Figure 3a). Nearly all P occurs between late autumn and winter (November to March), due to the circulation of cold air masses formed over the North Atlantic Ocean, in addition to deep pressure lows that move eastwards, resulting in an influx of air masses over the Subtropical Atlantic Ocean [39]. In late summer and autumn months, humid air masses formed over the western Mediterranean Sea may also generate P in eastern coastal regions of Spain, but these events seldom occur far inland [40]. Annual mean Ta varies between 4.6 °C in mountain ranges to 21.1 °C in low-lying large river basin locations (Figure 3b); the coldest and hottest months are January and August, respectively. In any given year, the daily Ta amplitude recorded in southern plateau and river valleys may achieve highs of 50 °C. The pronounced negative gradient of Ta in mountain ranges provides optimum conditions for producing seasonal snow-melt, credited as a principal source of freshwater for filling surface and GW bodies [41].

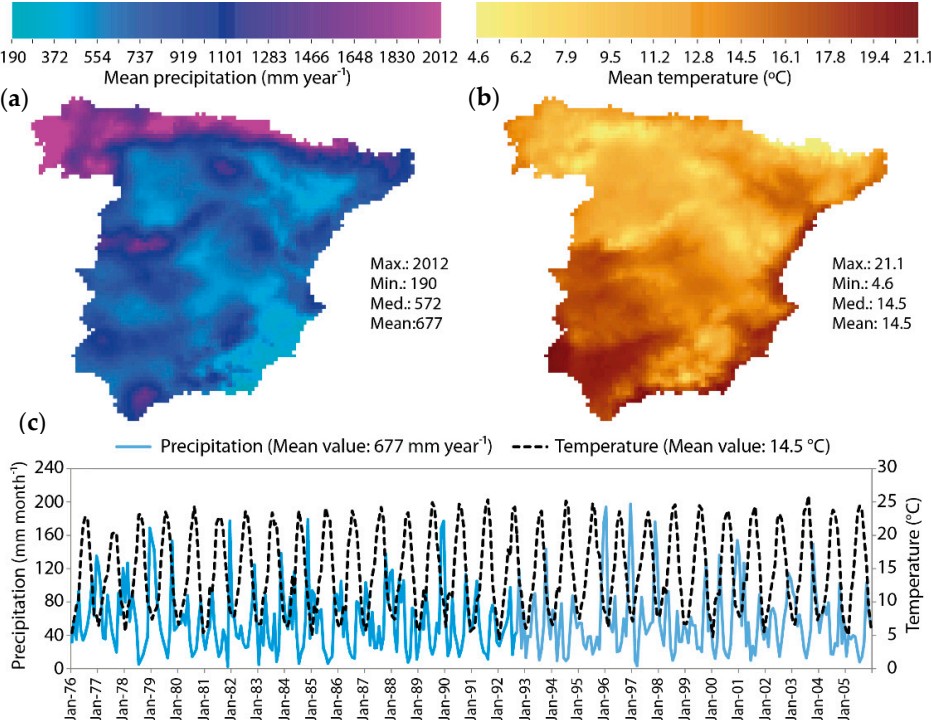

**Figure 3.** Map of historical mean (**a**) precipitation (mm year$^{-1}$) and (**b**) temperature (°C) across continental Spain during the reference period (1976–2005), (**c**) temporal series of mean precipitation (mm year$^{-1}$) Modified from [29].

2.2.2. Estimated Future Climatic Data

In order to generate future local scenarios, the historical climatic data (P and Ta series) in the reference period (1976–2005) were combined with the Climatic model simulations for the Control period (1976–2005) and future scenarios (2016–2045). In this study, we used data generated in previous works to assess the impacts of climate change on R in continental Spain [29]. This includes various climatic model simulations undertaken by the CORDEX EU project [42] for the most pessimistic IPCC emission scenario, the Representative Concentration Pathways 8.5 (RCP8.5). Selected simulations consist of results from five Regional Climate Models (RCMs) (CCLM4-8-17, RCA4, HIRHAM5, RACMO22E, and WRF331F) nested within four distinctive General Circulation Models. An equi-feasible ensemble of all RCM simulations was performed using 1976–2005 as the control/historical reference period and fixing the future horizon scenario as 2016–2045.

The RCPs are the greenhouse gas concentration trajectories adopted by the IPCC. They are named according to the radiative forcing that they represent. Radiative forcing is the change in the net downward minus upward radiative flux at the troposphere or top of the atmosphere due to a change in an external driver of climate change. The RCP8.5 is the most pessimistic pathway for which radiative forcing reaches values greater than 8.5 W m−2 by 2100. The selected RCM projections were performed using simulations of the RCP8.5 trajectories to generate potential future series of P and T. In this work, we corrected these series to generate local scenarios and to propagate their impacts on R.

The RCM climate modelling simulates climate conditions defined with some initial conditions, time-dependent lateral meteorological conditions and surface boundary conditions, to drive high-resolution models. These conditions are typically wind components, temperature, water vapor, and surface pressure. The driving data are derived from GCMs that simulate with a coarse resolution. Table 1 shows the GCMs used by the RCMs employed in this work. The World Climate Research Programme (WCRP) through the CORDEX project guarantees the quality of the RCMs they collected. However, uncertainties related to RCMs can be important and they must be adapted to the study area.

**Table 1.** Regional Climatic Models (RCMs) and General Circulation Models (GCMs) considered to define the climatic scenarios.

| GCMs / RCMs | CNRM-CM5 | EC-EARTH | MPI-ESM-LR | IPSL-CM5A-MR |
|---|---|---|---|---|
| CCLM4-8-17 | X | X | X | |
| RCA4 | X | X | X | |
| HIRHAM5 | | X | | |
| RACMO22E | | X | | |
| WRF331F | | | | X |

The monthly bias of the model within the reference period (1976–2005) was estimated as the mean relative differences between the control simulation and the historical P and Ta time series calculated for each month of an average year. This was used to generate the future series by applying a bias correction technique (scenario $E_B$). The monthly delta changes between control and future P (2016–2045) were also estimated to generate series by applying a delta change approach (scenario $E_D$) (Figure 4). Figure 4 shows that the potential future mean P and Ta generated for the scenarios $E_B$ and $E_D$ are exactly the same, although the temporal evolution of these variables is different, due to the different way in which they are generated from the historical values.

2.2.3. Potential GW Storage under the Surface Connection

For each of the selected GW bodies, we have taken the available information about potential GW storage under the surface connection (Figure 5). These were collected from the last River Basin Plans (2015–2021) published by the different River Basin authorities. This summarizes geological

and topographical information to define the GW body geometry, that combined with the storage coefficients provide the S (Mm$^3$) value for these GW bodies.

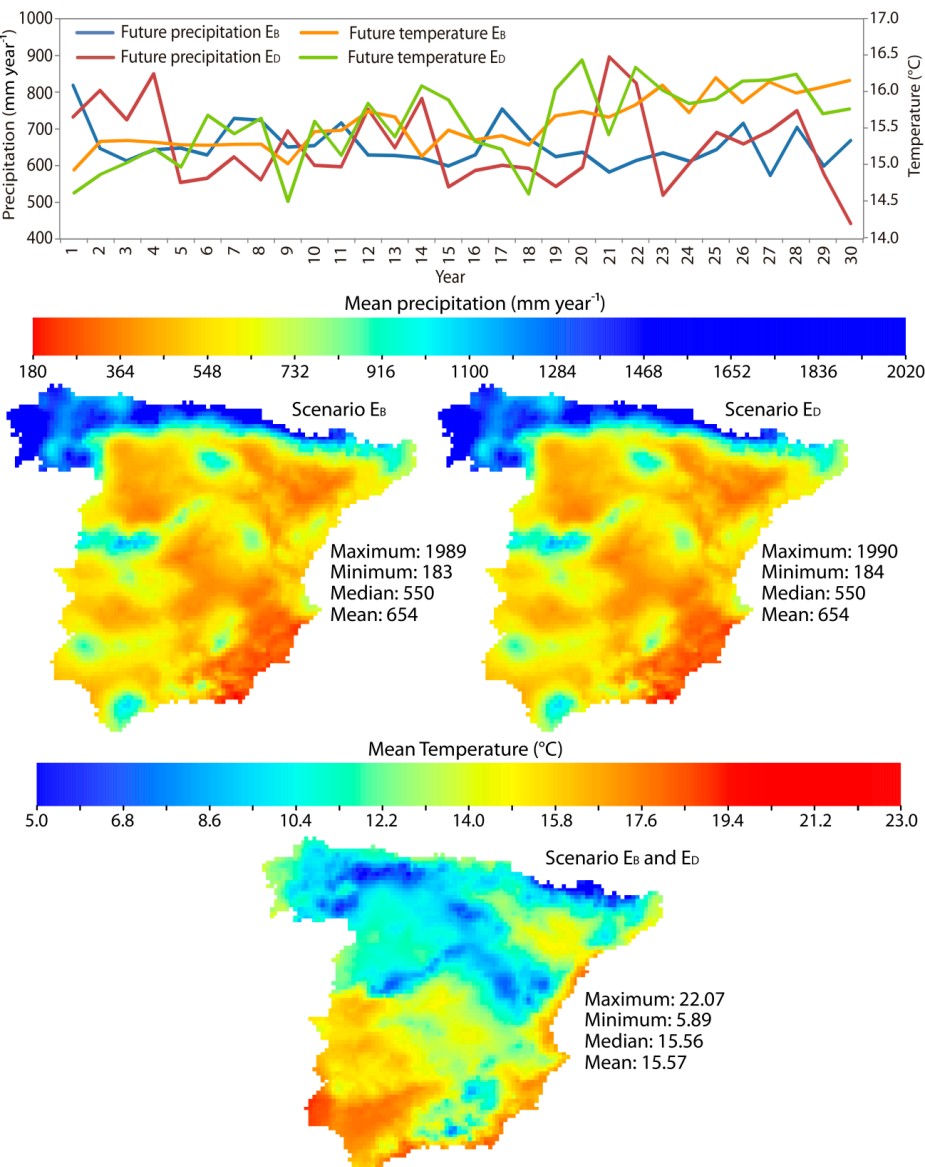

**Figure 4.** Potential future mean precipitation (mm year$^{-1}$) and temperature (°C) obtained with the equi-feasible delta and bias ensembles scenarios ($E_D$, $E_B$). Modified from [29].

### 2.2.4. Net GW Recharge: Historical and Future Scenarios

An empirical precipitation-R model was employed to estimate the historical R within the reference period and the impacts of potential future climatic scenarios on R [29]. It is defined as follows:

$$R = C(P - E) \tag{2}$$

where R, P, and E in mm year$^{-1}$ and dimensionless C are defined in caption of Figure 1. For estimating E, we used the non-global Turc [43,44] formulation:

$$E = \frac{P}{\sqrt{0.9 + \frac{P^2}{L^2}}} \tag{3}$$

where L = 300 + 25Ta + 0.05Ta$^3$ is a dimensionless form parameter of annual temperature.

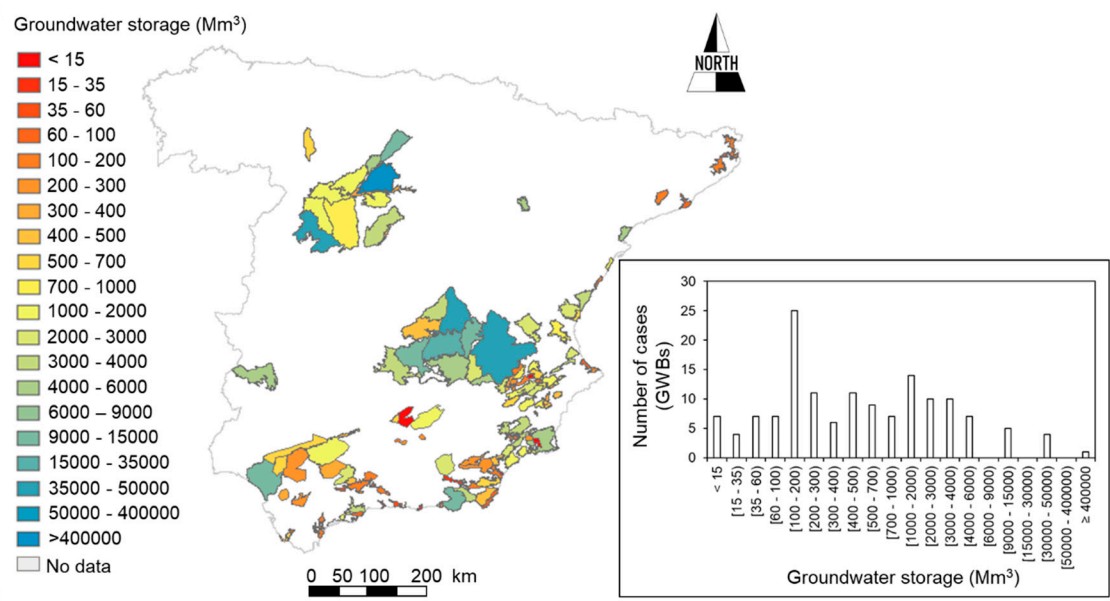

**Figure 5.** Potential storage capacity (Mm$^3$) under the surface connection for the 146 Spanish GW bodies at quantitative risk of not achieving the WFD [1] objectives.

This model has been used to propagate the impacts of local historical and future climatic fields in continental Spain (Figure 6).

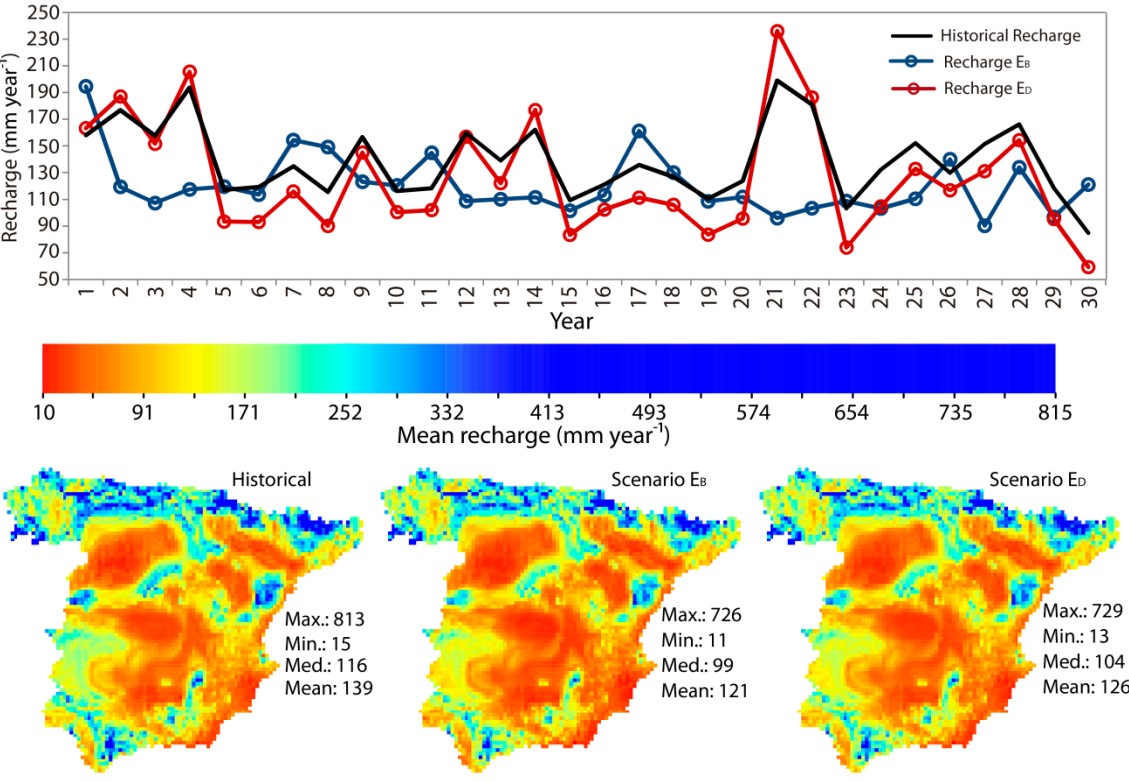

**Figure 6.** Historical (1976–2005) and future (2011–2045) potential R (mm year$^{-1}$) for the 2 defined equi-feasible ensemble scenarios. Modified from [29].

## 3. Results and Discussion

### 3.1. The T Index in Continental Spain: Historical and Future Scenarios

The information summarized in the previous section was used to assess the natural T for the historical period (reference period 1976–2005) and future potential scenarios in the horizon 2016–2045 that correspond to the RCP 8.5 emission scenario (Figure 7). Two different local climatic scenarios have been considered to assess the potential impacts on T values, one generated by an ensemble of bias correction approaches ($E_B$) and another by an ensemble of delta change approaches ($E_D$). The methodology and the series generated for those scenarios were described in Section 2.2.2. Figure 7 shows a heterogeneous distribution of T values within the 146 selected GW bodies as case studies. The box whiskers plot also reflects this wide range of T values moving from a minimum of 0.25 to a maximum of 3693 years in the historical period. The minimum and maximum values in the future scenarios are 0.32 and 4176 years for $E_B$, and 0.28 and 3953 years for $E_D$. In order to understand the variability of this value, we should take into account the formulation applied to estimate T (Equation (1)) in each GW body, defined as S divided by R, where S depends on the geometry ("the size" of the GW body) and the storage coefficients (hydrodynamic parameter depending on the geology and hydraulic behavior of the aquifer). Therefore, this variability in T values is logical taking into account the varied geology, size and hydraulic behavior of the considered GW bodies, as shown in Figure 2. The influence of S, and, therefore, the combined influence of geology, size and hydraulic behavior on T, is described in Section 3.2, where a sensitivity analyses of T values to S and R is performed. In addition, we also analyze the influence of the environmental conditions, taking into account that R depends on climatic and ground characteristics.

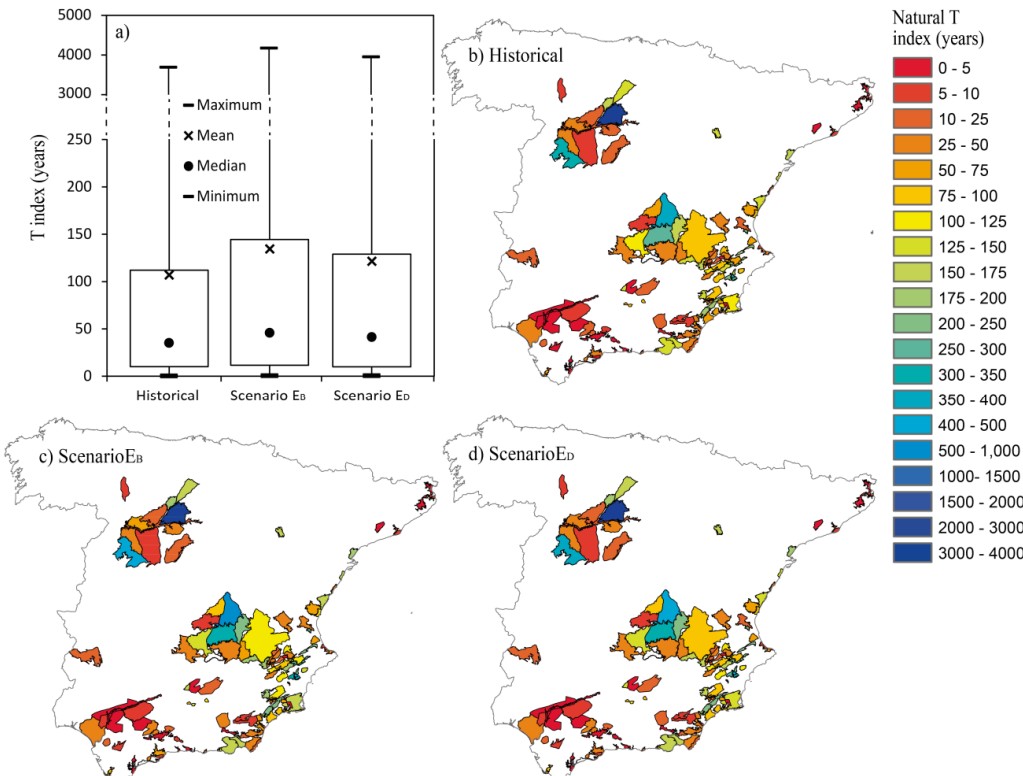

**Figure 7.** Box-whiskers (**a**) and maps of the T index in the 146 Spanish GW bodies at risk [1]. Historical (**b**) values and future potential scenarios (EB (**c**) and ED (**d**) in the horizon 2011–2045. The differences between the future scenarios ($E_B$ and $E_D$) in terms of impacts on the T index are small, due to the differences between the impacts on mean R also being small (see maps of Figure 6). The mean values of R for both scenarios are very similar, although the monthly series are different (see temporal series of Figure 6).

Low T values means that R is close to S, and therefore, they are extremely vulnerable to exploitation, even in periods when pumping is smaller than the average R. This can be especially relevant in areas with scarce resources where long and intensive drought appear and will be exacerbated in the future due to climate change. If we assume that the long term management of the Water Resource Systems allows to maintain the natural mean reserves (the mean S) of the GW bodies, the highest values of T correspond to GW bodies that can be very useful due to their buffer values role in managing drought periods. Around 26.9% of the studied GW bodies show low pumping vulnerability with historical T values above 100 years, with this percentage increasing to 33.1% in the near future horizon values (until 2045).

Taking into account the formulation employed to assess T as S divided by R (see Equation (1)), the impacts of the future scenarios on T are explained by the change in R, which is the only variable that depends on the climatic conditions. The T values will increase in the future in most of the GW bodies (Figure 8) due to the recharge (R) being reduced; meanwhile the total potential storage under the surface connection (S) will stay invariant. The impacts of potential future scenarios on T values will be heterogeneous (see maps of Figure 8). The box whiskers plot also reflects a wide range of T value changes with respect to the historical values moving from a reduction of 2.8 years to an increment of 483 years, which is due also to the variability observed for the recharge, where we estimate changes with respect to the historical between a reduction 47.0 mm year$^{-1}$ and an increment of 2.7 mm year$^{-1}$.

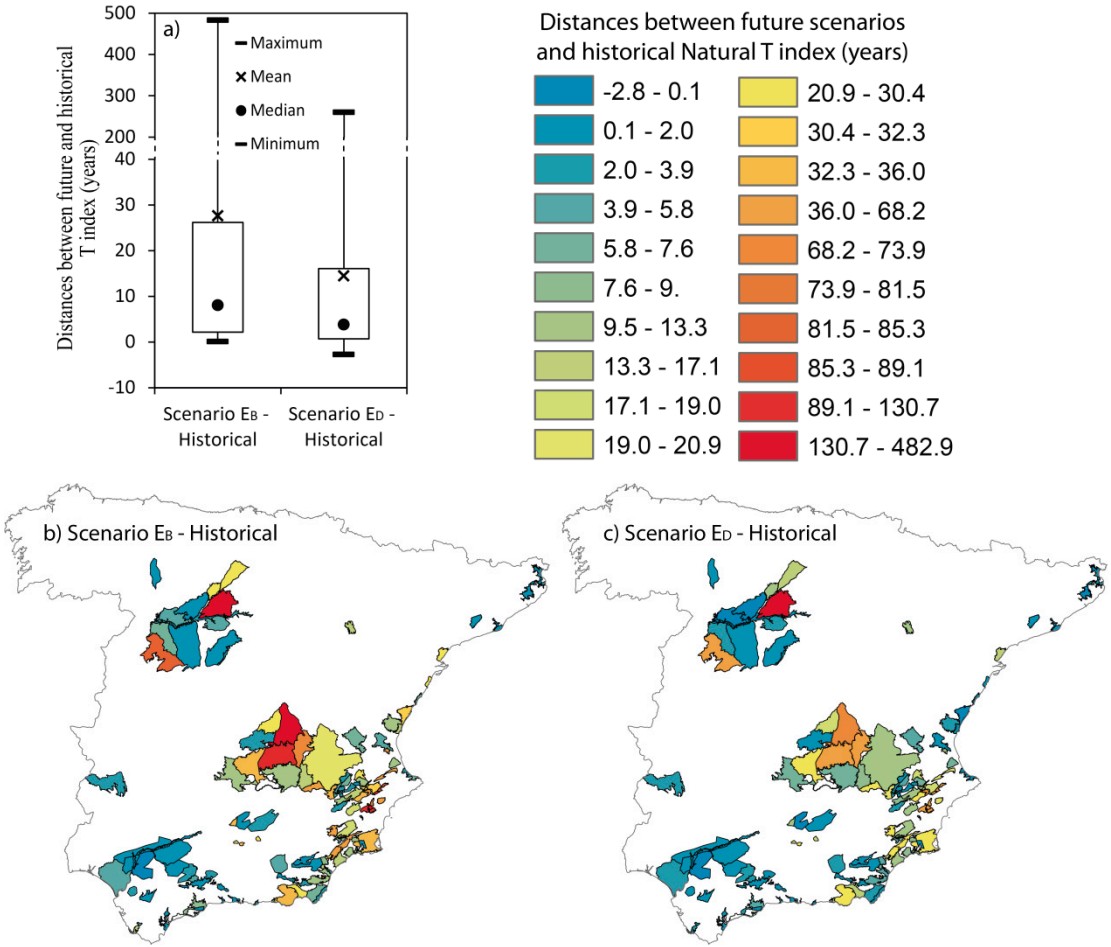

**Figure 8.** Box-Whiskers (**a**) and maps (**b**,**c**) of the distances between historical natural T and future potential values in horizon 2011–2045.

The increments in T values will force the application of more restrictive long-term management strategies within the systems to maintain the natural mean reserves, but if this long term constraint

is fulfilled, the potentiality of those GW bodies to be used to play a buffer role to manage drought periods will be in many cases even higher than in the historical period (Figure 8).

### 3.2. Influence S and R on the T Index for Different Lythologies

The T–S and T–R data-pairs were compared attending to the predominant lithology of each GW body at risk. In general, for a given lithology, higher correlation is observed for T–S values (Figure 9a) than for T–R (Figure 9b). This can be observed more clearly in carbonated GW bodies ($R^2 = 0.82$ for T–S relationship and 0.04 for T–R), while in the detrital GW bodies $R^2$ would be 0.47 and 0.15, respectively. This could be explained by the lower variation of R in the carbonated GW bodies (see Figure 9). Nevertheless, in general the variability of the R values (ranging between 22.8 and 309.7 mm year$^{-1}$) is significantly smaller than the S values (ranging between 2.9 and 401,1875 Mm$^3$), which is the variable that better explains the high variability of T values (ranging between 0.3 and 3693.2 years), especially in carbonated GW bodies, where the variability of R is very low.

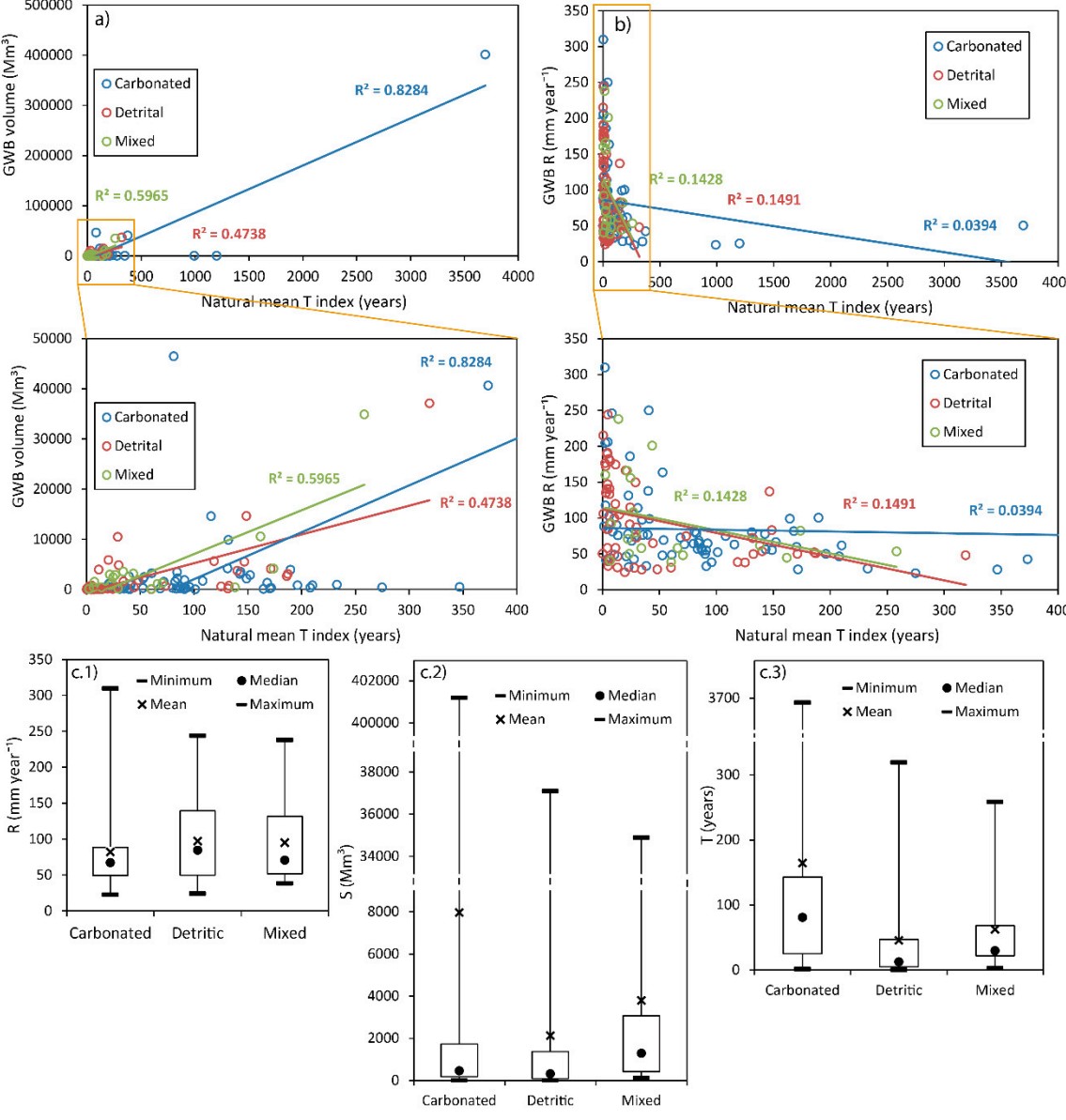

**Figure 9.** (**a**) GW body volume (Mm$^3$) vs. natural mean T index (year), (**b**) R (mm year$^{-1}$) vs. natural mean T index (year), T and (**c**) box-whiskers of R, S and T for the three considered GW bodies lithological categories: carbonated, detrital, and mixed.

This analysis has also been extended to other explanatory variables, such as the "recharge coefficients" defined as RC = R/P (See Figure 10b); and the effective recharge coefficients C = R/(P-E) (see Figure 10c and Equation (2)). This work assumes that C is a parameter that will stay invariant when assessing future impacts on GW bodies R, which is a common assumption performed in future projection studies [35,45], despite that, theoretically, they may vary due to changes in vegetation cover, land use, soil properties, and structure of rainfall events, as foreseen by the Spanish desertification model prepared in the framework of the National Plan to Combat Desertification [46,47].

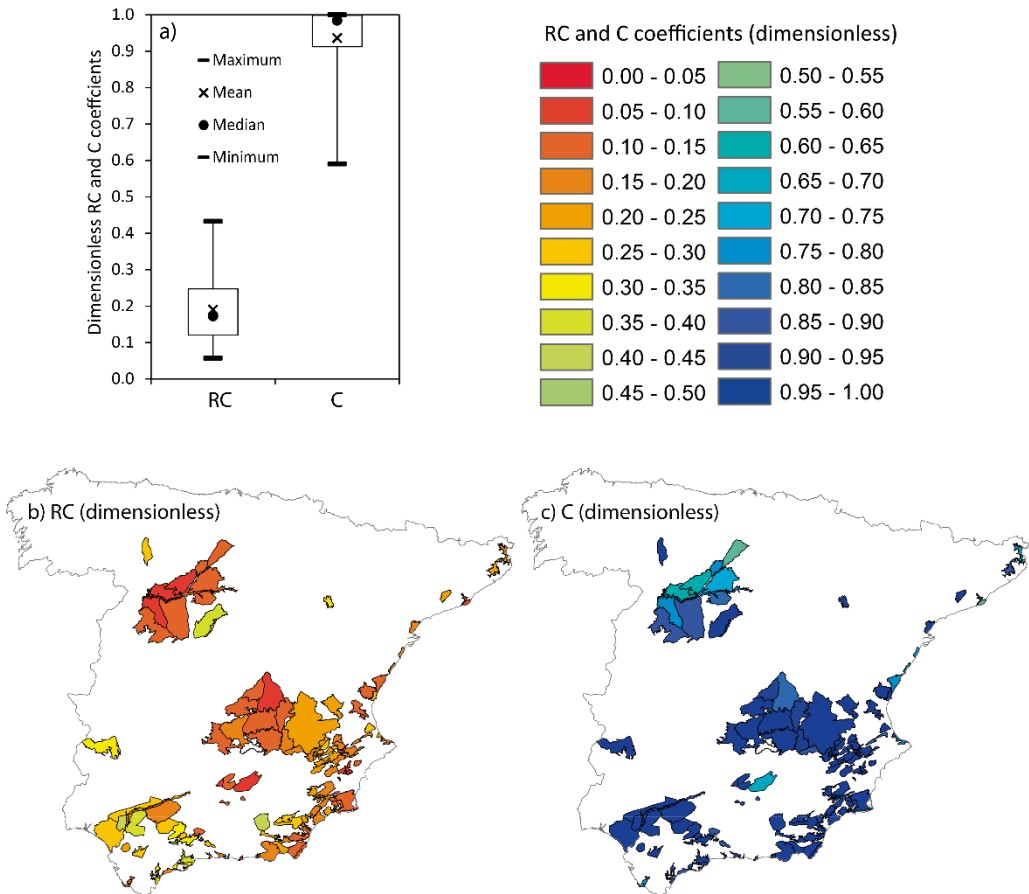

**Figure 10.** Box-Whisker (**a**) and maps of the of the "recharge coefficients" (RC) (**b**) and the "effective recharge coefficient (C)" (**c**) in the GW bodies at risk [1].

The difference between C and RC gives an idea of the impacts of the actual evapotranspiration in the calculation of recharge. Higher differences indicate longer distance between precipitation and effective precipitation, defined as the precipitation minus the actual evapotranspiration. We found that the T changes (future scenarios vs. historical period) are higher when the difference between C and RC are higher (see Figure 11). Therefore, the GW bodies with higher difference between C and RC are more sensitive to climate change.

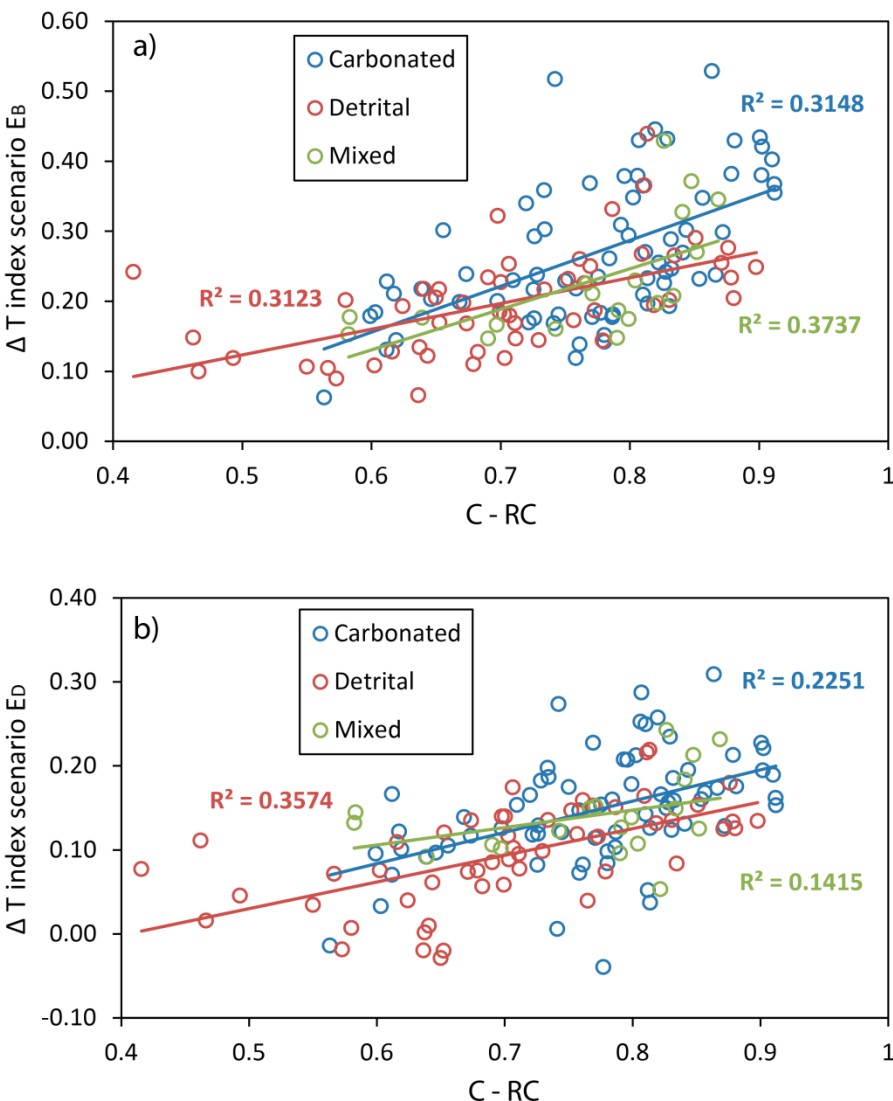

**Figure 11.** Absolute distance (years) of future T values (**a**) EB scenario and (**b**) ED scenario, with respect to the historical T vs. difference between effective recharge coefficients and recharge coefficients.

### 3.3. Hypothesis Assumed and Limitations of the Method (Recharge and Total Storage Uncertainty)

In order to estimate the natural mean GW renewable time, we assume that there are no changes in Land Use and Land Cover (LULC), no pumping, and the net GW discharge will start when the potential total GW storage reaches the level of the surface connection (spring, streambed, river level, or sea level boundary condition) (See Figure 1). As described, a parallelism between unconfined aquifers and reservoirs is adopted to approach this by the T index. Analogies between an unconfined aquifer and a reservoir have been previously adopted to approach different stream-aquifer interaction problems [48–51] The T values are used to assess vulnerability to pumping during droughts, assuming that the LULC and the management of the water resources system will allow the natural mean reserves (the mean S) of the GW bodies to be maintained, even under long term future R scenarios. Under this assumption, highest values of T correspond to GW bodies that can be useful to manage droughts due to their buffer values role.

We assume that the historical S values derived from the last River Basin Plans published by the different River Basin authorities (2015–2021) are good enough for a preliminary approximation of the T values.

The historical R in the reference period (1976–2005) has been estimated by applying an empirical approach [29] with a spatial resolution of 10 km × 10 km, which is considered an accurate enough approach for a preliminary simple assessment of T. The R model is based on a preliminary approach to the main drivers of the R dynamic (precipitation, actual evapotranspiration and "effective recharge coefficients") in accordance with the available data and, although it is not a state of the art model, it produces a good enough preliminary approach to identify potential strategic GW bodies, where more detailed studies will be required for a more accurate assessment. The main hypothesis and limitations of the proposed approach are:

- The climatic fields (P and Ta) in the case studies are approximated by the Spain02 project dataset [38]. This dataset has been recently validated by Quintana-Seguí et al. [52] and has already been employed in many research studies.
- We assume that the mean yearly long term E assessment provided by the non-global Turc's model [43,44], whose results depend on mean annual Ta and P, is good enough for this preliminary assessment. Due to its simplicity and efficiency this approach has been extensively applied in research works in which preliminary E assessment for historical and/or future scenarios is included [33,53]. In spite of this, more accurate assessments of groundwater resource will require, in addition to the non-global E approach, corrections by using global models for E [54] or an external calibration by using well-suited recharge functions [55]. This is especially interesting in the Spanish drylands, where E is typically close to P [53,54]. In this work we use the second option, i.e., the Turc formulation. and the "effective recharge coefficients" deduced from a previous calibrated recharge function provided by Alcalá and Custodio [31,32].
- The "effective recharge coefficients" (C) are calibrated from the R values derived by Alcalá and Custodio, [31,32]. They used a chloride mass balance method whose accuracy is similar to that obtained when global models for E are used for recharge purposes [55]. These values are available at a spatial resolution of 10 km x 10 km grid, and therefore the R model will also be at this scale. We assume that the future impacts of potential climatic scenarios on R can be obtained by propagating future local scenarios with the R model previously calibrated. Therefore, this assumes that the effective recharge coefficient remains invariant in simulating future conditions.
- The analyses of future potential climatic scenarios do not include the simulation of any future LULC scenarios and/or management scenarios of the water resources system. We only analyze potential impacts on T, and, therefore, on R, due to climate drivers, assuming that, in the future, the LULC and management will allow the maintenance of long term mean natural reserves. Under this assumption, the proposed method will be useful in the future to identify strategic resources to manage droughts. In the literature we found several research works in which the potential future impacts on aquifers are analyzed taking into account only climate drivers (Pulido-Velazquez et al., 2018; Pardo-Iguzquiza et al., 2019). The development of additional research works will be needed to study specific LULC and management issues (Pulido-Velazquez et al., 2018).

Future local climate driven scenarios have been generated for a short-term horizon (2015–2045) assuming the most pessimistic emission scenarios RCP8.5.

- Local projections have been obtained from different climatic model simulations by applying two downscaling approaches (correction of first and second order moments) under two different hypotheses (bias correction and delta change techniques) [33,35].
- The final scenarios employed to study potential impacts on R and T have been defined by an equi-feasible ensemble of local projections, which produce more robust and representative projections than those based on a single model [36].

## 4. Conclusions

Aquifers with higher GW mean residence time show lower vulnerability to pumping during drought periods. T values are used to assess vulnerability to pumping during drought periods, assuming that the long-term management of the Water Resources Systems will allow for the maintenance of natural mean reserves (the mean S) of the GW bodies, even under future long-term recharge (R) scenarios. A preliminary assessment of this variable can be obtained by the natural mean turnover time (T) index defined as S divided by R. Aquifers where R is close to S are extremely vulnerable to exploitation, even in periods when pumping is smaller than the average R. This can be especially relevant in areas with scarce resources where long and intensive droughts appear and will be exacerbated in the future due to climate change. In this work we identify potential strategic GW resources, with low vulnerability to pumping, which can be useful to define sustainable conjunctive use management of droughts in continental Spain. We focus our analyses on the Spanish GW bodies at risk of not achieving the European Water Framework Directive [14] objectives (146 GW bodies). We performed a historical and future (short term period until 2045) assessment of T as the S/R ratio. Around 26.9% of these GW bodies show low pumping vulnerability with historical T values above 100 years, with this percentage increasing to 33.1% in the near future horizon values (until 2045). The results observed in the study area show a significant heterogeneity. The maximum range of the historical T variability is around 3700 years, which also increases in the near future to 4200 years. Therefore, the vulnerability to pumping is also quite heterogeneous. The T index values will change in future horizons, and the potential use and the constraints to be applied in using GW bodies to define conjunctive use strategies in order to adapt to Climate Change scenarios will also change in the coming years. We have also analyzed the variability and influence of R and S values in the determination of T for different aquifer lithologies.

**Author Contributions:** D.P.-V. planned the research/methodology, and contribute to writing and reviewing the manuscript. J.R.: contribution to the assessments, figures and writing activities. A.-J.C.-L.: contribution to the assessments, figures and writing activities. F.J.A.: contribution to writing and reviewing the manuscript. F.F.-C.: contribution to the assessments. L.B.-R.: contribution in figures and writing activities. Valuable comments and suggestions were provided by four anonymous referees. All authors have read and agreed to the published version of the manuscript.

**Funding:** This research has been partially funded by the projects GeoE.171.008-TACTIC and GeoE.171.008-HOVER, funded by European Union's Horizon 2020 research and innovation program; and SIGLO-AN (RTI2018-101397 -B-I00) project from the Spanish Ministry of Science, Innovation and Universities (Programa Estatal de I+D+I orientada a los Retos de la Sociedad).

**Acknowledgments:** We thanks the reviewers and the editor for the valuable comments they provided to improve the manuscript.

**Conflicts of Interest:** The authors declare that they have no known competing financial interests or personal relationships that could have appeared to influence the work reported in this paper.

## Abbreviations

| | |
|---|---|
| S | Groundwater storage |
| R | Net groundwater recharge |
| T | Natural mean groundwater turnover time |
| GW | Groundwater |
| WFD | European Water Framework Directive |
| E | Actual evapotranspiration |
| P | Precipitation |
| Ta | Temperature |
| Q | Net groundwater discharge |
| U | Groundwater pumping |
| RC | Recharge coefficient or P-to-R ratio |
| C | Effective recharge coefficient |

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
