# Peer review of "Using the Turnover Time Index to Identify Potential Strategic Groundwater Resources to Manage Droughts within Continental Spain"

_water, doi:10.3390/w12113281_

Round 1
Reviewer 1 Report
The authors use an index defined by the storage capacity of the aquifer divided by the recharge. The model use to characterize the recharge is not state of the art: the evaporation is estimated by the Turc formula and the recharge is proportional to the difference between precipitation and evaporation. More reliable approaches exist. Moreover, soil cover and agricultural practices will change in the near future which is not taken into account or discussed in details.
Therefore, the manuscript cannot be published in its present form.
Author Response
We have carefully analyzed the reviewers’ comments. Thanks to them, we have realized that some aspects of the original manuscript were not too clearly presented, and we have made an effort to conscientiously improve the clarity of them.
Detailed responses are provided to the reviewers’ comments in the response_Reviewer Comments file.

Reviewer 2 Report
General comments:
- This is a useful study addressing the need for proper management of droughts throughout the world in response to the potential impact of climate change on fresh water resources. However, the authors should provide a better discussion of the results, and present the reasons behind the findings of this study.
- The manuscript contains grammatical errors that should be corrected. The manuscript should be read and corrected by a native English speaker.
- The manuscript does not contain line numbers which made the review process difficult.
- The section numbers should be checked carefully and corrections need to be made.
Specific comments:
- Page 2/19, second paragraph: WFD should be spelled out at first use.
- Section 2.2.1. Location, Geological Context and Historical Climatic Data: The type of parameters and the respective values of parameters that were used to identify and select 146 Spanish GW bodies in risk of not fulfilling the WFD (2000) objectives should be described in the manuscript.
- Section 2.2.1. Estimated Future Climatic Data: Please briefly describe the most pessimistic IPCC emission scenario, the Representative Concentration Pathways 8.5 (RCP8.5) and its impact on the parameters used in this study, particularly P and R.
- Section 2.2.1. Estimated Future Climatic Data: The authors should also briefly describe the five Regional Climate Models (RCMs) (CCLM4-8-17, RCA4, HIRHAM5, RACMO22E, and WRF331F), the parameters used in these models, the strength and weaknesses of models and the quantitative values of parameters that were used in the simulation studies based on these models. What are the input and output parameters in these models?
Results and Discussion:
- Section 3.1. The T Index in Continental Spain: Historical and Future Scenarios: EB and ED should be described.
- Section 3.1: The authors should demonstrate and discuss how the variations in geology, size and hydraulic behavior of the considered GW bodies impact the T values, and show how the wide range of T values obtained in the simulation studies can be justified. Additionally, the impact of environmental conditions at the sites examined in this study on the T values should be discussed.
- Section 3.1: The authors state that the impacts of potential future scenarios on T values will be very heterogeneous (maps of Figure 8). The reason behind this observation should be explained and the type of parameters as well as their variability that control the heterogeneity of future scenarios should be thoroughly described.
- Page 14/19: The definitions of precipitation and effective precipitation, as applied in this study, should be presented.
Author Response

(The authors gave the same response as above.)

Reviewer 3 Report
- Figure 1: What does “ &E “ mean?
- Figure 3: Please add the explanation of the figure “c”.
- Figure7: What is the difference of Scenarios EB and ED?
Author Response

(The authors gave the same response as above.)

Reviewer 4 Report
The work proposed by Velazquez et al entitled “Using the Turnover Time Index to Identify Strategic Groundwater Resources to Manage Droughts within Continental Spain” aim to assess the intrinsic aquifer vulnerability to pumping stress in the next future (2045). The work is well written and interesting, only a check of error is necessary in some part of the text. I think the work is suitable for publication, here my minor comments:
The Abstract reflect more a conclusion paragraph than an abstract itself, I suggest to rewrite it considering the classical structure. In few word in necessary to introduce the problematic, propose the innovation or the aim and illustrate the results.
“GW overexploitation is an issue even with higher impacts in lowering GW levels than climate change in many regions, especially in the Mediterranean area [7]”. Please discuss the issue more in detail proposing some examples.
“The concept of vulnerability is closely related to the GW body status and risks. It has been extensively studied from the perspective of vulnerability to surface pollution. From this qualitative point of view, the vulnerability of a GW-resource to pollution depends on intrinsic susceptibility, which depends on the aquifer properties and the associated sources of water and the distribution and types of contamination sources (natural and/or anthropogenic), and the transport of the contaminants [8]. This aquifer vulnerability concept has been also linked to the variables GW ages, travel and residence time” Also in this case I suggest to highlight the more recent funding in literature that have contribute in the evolving of the GW-vulnerability concept. Here a suggestion to discuss:
Busico, G., Kazakis, N., Cuoco, E., Colombani, N., Tedesco, D., Voudouris, K., & Mastrocicco, M. (2020). A novel hybrid method of specific vulnerability to anthropogenic pollution using multivariate statistical and regression analyses. Water Research, 171
Machiwal, D., Cloutier, V., Güler, C., & Kazakis, N. (2018). A review of GIS-integrated statistical techniques for groundwater quality evaluation and protection. Environmental Earth Sciences, 77(19)
Author Response

(The authors gave the same response as above.)

Round 2
Reviewer 1 Report
The authors provided a significantly improved revised manuscript. The answers to most of the reviewers comment are relevant.
They considered that my comments on the estimation of evapotranspiration and changes in land use and agricultural practices are out of scope.
I do not agree but I leave the final decision to the editor to publish the paper as it is. Since I am obliged to select an overall recommendation, I recommend major revision.
Reviewer 2 Report
The manuscript has much improved after the revision. I have no more comments and recommend this manuscript for publication.
